# Wine Closure Performance of Three Common Closure Types: Chemical and Sensory Impact on a Sauvignon Blanc Wine

**DOI:** 10.3390/molecules27185881

**Published:** 2022-09-10

**Authors:** Annegret Cantu, Jillian Guernsey, Mauri Anderson, Shelley Blozis, Rebecca Bleibaum, Danielle Cyrot, Andrew L. Waterhouse

**Affiliations:** 1Department of Viticulture and Enology, University of California Davis, One Shields Avenue, Davis, CA 95616, USA; 2Department of Psychology, University of California Davis, One Shields Avenue, Davis, CA 95616, USA; 3Dragonfly SCI, Inc., 2360 Mendocino Avenue, Ste. A2-375, Santa Rosa, CA 95403, USA; 4Cade Estate Winery, 360 Howell Mountain Rd S, Angwin, CA 94508, USA

**Keywords:** wine aging, browning, closure, spectroscopy, QDA^TM^

## Abstract

A Napa Valley Sauvignon blanc wine was bottled with 200 each of a natural cork, a screw cap, and a synthetic cork. As browning is an index for wine oxidation, we assessed the brown color of each bottle with a spectrophotometer over 30 months. A random-effects regression model for longitudinal data on all bottles and closure groups found a browning growth trajectory for each closure group. Changes in the wine’s browning behavior at 18 months and 30 months showed that the browning of the wine bottles appeared to slow down later in the storage period, especially for natural corks. The between-bottle variation was the highest for the natural cork. At 30 months, we separated the bottles by the extent of browning and samples were pulled from the high, mid, and low levels of browning levels for each closure. The degree of browning is inversely correlated with free SO_2_ levels ranging from 5 to 12 mg/L. However, a Quantitative Descriptive Analysis (QDA™) sensory panel could not detect any difference in their aroma and flavor profile between closure types regardless of browning level. Even low levels of free SO_2_ retain protection against strong oxidation aromas, and visual browning detected by spectrophotometer seemed to precede oxidative aroma and flavor changes of the aging Sauvignon blanc.

## 1. Introduction

A wide variety of closures are available on the market. Cylindrical closures made of natural cork, technical cork, or synthetic material, and screw caps with liners inserted are the major closures used today in the U.S. wine industry [1,2,3].

For the winemaker, closure choice is a critical winemaking decision, as closures are known to influence the wine post-bottling due to their performance. The main functions of closures are to seal and prevent liquid leakage during storage. However, one of the closure’s key characteristics and performance concerns is permeability to oxygen and its resulting impact on the wine’s sensory profile. In the past decades, several wine-closure aging studies have been addressing this topic with different approaches [4,5,6,7,8,9,10,11,12,13,14]. Oxygen, as already described by Louis Pasteur over a century ago, is both an enemy and friend to wine quality that can alter the final product significantly during aging [15]. It can contribute to flavor development and help with color stability and astringency reduction in red wine [14,16,17]. Yet, too much oxygen in white wines will quickly cause wine deterioration, e.g., browning, and premature aging [12,18]. Likewise, too little oxygen in wines that produce low molecular weight sulfur compounds (e.g., H_2_S, MeSH) during aging can lead to suppression of some flavor compounds, as well as struck flint and rubbery tasting notes, also known as ‘reductive’ aromas [12,19,20].

The closure’s barrier function to oxygen and its consistency has been established in the literature by determining the oxygen transfer rate (OTR) [21,22,23]. OTR is, by definition, the amount of oxygen that permeates through the closure at a steady state. The oxygen permeability of closures has also been measured by several methods reported in the literature as oxygen ingress [10,13,24,25] using model wine solutions and dye titrations. Aside from the closure regulating the oxygen ingress after bottling, total package oxygen (TPO) also needs to be considered. This is oxygen present at bottling in the headspace, in the wine, and the closure itself. Oxygen entrapped in the closure is released over a few months after bottling [9,10,26]. Therefore, a closure’s oxygen barrier function plus TPO can impact chemical oxidation and affect the quality, product consistency, and life span of a wine [10,13,27,28]. A prior study on a Semillon wine clearly demonstrated that closures with different material properties did not have the same oxygen barrier functions. Consequently, the variety of closures investigated created different wines after an extended aging period [5,19].

Moreover, a closure study demonstrated that higher oxygen ingress and TPO correlated to free sulfur dioxide decline, an increase in wine browning, and, as a result, quality decline [5,8,19,28]. The same Godden et al., 2001 study also found inconsistencies in closure performance within the same type, e.g., variations due to TCA tainted corks. Others found considerable variability in oxygen permeability in natural cork and in synthetic [13] and screwcap closures with different liner materials [24]. These studies speculated that closures with high variability within lots could lead to wine bottles of large-quality differences. However, in other studies, manufactured closures like technical corks, synthetics, and screw caps have shown high homogeneity [12,23].

In summary, several publications reported closure inconsistency related to oxygen barrier properties [10,13,19,23,24]. Nevertheless, the actual sensory and chemical impact on wine was not studied in a definitive manner.

This study aimed to investigate the variability within wine bottles using a specific closure, and to test whether consumers could detect and describe the differences in sensory characteristics of two “identical” bottles of the same closure lot after an extended aging period with closures of differing OTR. Moreover, we wanted to explore the chemical differences between those same bottles of wine within and among its closure groups. For this purpose, a “fruity and crisp” Sauvignon blanc wine was bottled with three commonly applied closures. The browning of each bottle at A420, a proxy for oxidation, over 30 months was assessed to follow the contribution of each closure’s consistency to an ‘aging’ browning OTR trajectory’. Furthermore, at the end of the experiment, we measured basic oxidation-related wine chemistry. We undertook a QDA sensory analysis to investigate whether non-expert tasters could detect any differences within and among closures, comparing the most and least oxidized bottles.

## 2. Results

### 2.1. Browning Trajectories of a Sauvignon Blanc Wine under Three Different Closures

The overall browning behavior of the wine bottles for each closure type is shown in fitted browning trajectories that will be further elucidated (Figure 1). When comparing the different growth models, there was no evidence of variability in the linear rate parameter for the linear growth model or in the linear and quadratic rate parameters for the quadratic growth model for all closures. The so-called random effects were subsequently excluded from the models. Table 1 reports indices of model fit for the three growth models (see Appendix A), each of which includes only a random intercept to account for the clustering of scores within bottles. Browning levels for all three closures were best summarized by a quadratic growth function with a random intercept, as evidenced by this model having the lowest indices of model fit. Maximum likelihood estimates of model parameters based on the quadratic growth model with time centered at 18 months and 30 months are provided in Table 2. The plots of the estimated browning trajectories for the typical bottle by closure, plotted along with the sample mean for each closure, are given in Figure 1. The browning levels increased with time at a non-constant rate.

At 18 months, the estimated mean browning level was 0.79 (se = 0.006) for synthetic cork, 0.78 (se = 0.005) for natural cork, and 0.83 (se = 0.005) for screw cap. Between closures, the estimated browning levels differed on average between natural cork and screw cap (difference = 0.051, se = 0.007, t = 7.03, *p* < 0.0001) and between synthetic cork and screw cap (difference = 0.042, se = 0.008, t = 5.34, *p* < 0.0001). At 30 months, the estimated mean browning level was 0.84 (se = 0.006) for synthetic cork, 0.81 (se = 0.005) for natural cork, and 0.89 (se = 0.005) for screw cap. Between closures, the estimated browning levels differed on average between natural cork and screw cap (difference = 0.087, se = 0.007, t = 11.59, *p* < 0.0001), between natural cork and synthetic cork (difference = 0.031, se = 0.008, t = 3.75, *p* = 0.0002), and between synthetic and screw cap (difference = 0.056, se = 0.008, t = 6.98, *p* < 0.0001).

The mean browning levels calculated with the quadratic growth function at 18 or 30 months showed that synthetic closures and natural corks were very similar in browning trajectory curve progression (Figure 1). However, the screwcap wines had higher browning differentials. Contemplating that, the browning growth curves of the screwcap and synthetic corks were very similar, while natural corks showed a deceleration in browning. The differences in their browning trajectories are well identified by respective browning acceleration rates (Table 2), where the most negative value for β_2_ shows the most considerable change in browning rate over time. Screw caps have the smallest change (β_2_ = −0.00005), followed by synthetic corks (β_2_ = −0.00008), while natural corks exhibit more significant slope change (β_2_ = −0.00013).

After accounting for growth in browning over time by the quadratic model that includes a random intercept to address between-bottle variation, the estimated variances that represent within-bottle variation over time are all large relative to their respective standard errors: For synthetic, σ^^2^ = 0.0003 (se = 0.00002); for natural, σ^^2^ = 0.0009 (se = 0.00004); and for screw cap, σ^^2^ = 0.0002 (se = 1.0 × 10^5^). A log-linear model [29] is used to test for differences in the estimated variances between closures. The estimated within-bottle variance for natural cork is greater than that for both synthetic (t = 15.57, *p* < 0.0001) and screw cap (t = 22.18, *p* < 0.0001) and is greater for synthetic relative to screw cap (t = 6.58, *p* < 0.0001). 

### 2.2. Results for Degree of Browning, Free and Total Sulfur Dioxide Concentrations and QDA Sensory Analysis

For purposes of selecting bottles for sensory analysis, after 30 months, each single wine bottle was evaluated with the spectrophotometer for their total degree of browning. Wine bottles were evaluated for their individual linear slope of absorbance (ΔAU/day) including all the time points aside from the initial measurement. Further, we selected five bottles at the low, medium, and high end of the browning scale for sensory analysis to create three categories, high, medium, and low browning (Table 3). The linear slope-based browning levels were significantly different in these three categories within each closure group at *p* < 0.05 (Figure 2A). The average of these slope values was 131 ± 69 of log2 transformed absorbance units per day for the natural closures, 190 ± 28 for the synthetic, and 170 ± 28 for the screwcaps, showing overall a slower browning for natural corks on average, but with a higher standard deviation for the natural corks (Figure 2A).

When opening the bottles for the sensory panel, the free SO_2_ values measured at 30 months were significantly different within closure browning groups (Figure 2B). For screw cap (CL, CM, and CH), low (CL) and medium (CM) browning levels were not different but both were significantly different compared to the high browning level (CH) with respect to free SO_2_. The natural corks (NL, NM, and NH) followed the same pattern for the least-significant differences, but interestingly the high browning level (NH) had lower free SO_2_ than high browning level screw cap (CH) and high browning level synthetic cork (SH). The synthetic cork browning levels (SL, SM, and SH) were the same for the medium and high browning levels, in this group the low browning level had significantly higher free SO_2_ concentration.

The total SO_2_ levels (Figure 2C) for screw cap and natural cork closures were not significantly different for low (CL, NL) and medium (CM, NM) browning levels, only the higher browning groups (CH and NH) had significantly lower total SO_2_. For synthetic closures, no significant differences were detected between the three browning levels (SL, SM, and SH). Overall, there was a relationship between browning “slope” and the free and total SO_2_ remaining at 30 months (Figure 2A–C). The bottles with the high level of browning had the lowest level of remaining free SO_2_, with the low-browning bottles having high SO_2_.

It is interesting to note that among all closures, the natural cork closures with low (NL) and medium (NM) browning levels had the highest free and total SO_2_ concentrations after 30 months of aging.

We were interested in looking into the overall variation of browning for all bottles within each closure type. Figure 3 shows μ-absorbance value data (slope values/day) at 30 months of the three closures with scattergrams to visualize the variation in each closure. The natural cork browning data had a CV of 44.8% (green, left side), and the screw cap (orange, middle) and synthetic corks browning data (grey, on the right) had CV% of 20.6 and 21.9%, respectively.

The red crosses show the mean values and the red horizontal bars the median, and they indicate that natural cork closures had slightly lower average browning but more variation.

QDA sensory methodology is known to provide quantitative maps that allow for a detailed understanding of product similarities and differences within and between closure types. The panel, as a group, agreed on sensory terms, divided into five modalities to describe the wines: appearance (4), aroma (6), flavor/taste (9), mouthfeel (4), and aftertaste/aftereffects (10) (Appendix B for Panel QDA definitions of sensory terms). The appearance terms were golden color, green, thickness, and bubbles. The panel consented to oaky, alcohol, sour, fruity, sweet, and spice for aroma. For flavor/taste, the terms were green apple, fruity, citrus, oaky, buttery, alcohol, bitter, sweet, and sour. For mouthfeel/texture, puckering, tingling, burning, and vapors in the back of the throat were rated. Aftertaste and aftereffects were rated as green apple, fruity, alcohol, citrus, sour, bitter, oaky, dry, burning, and mouthcoating. For these 33 consent sensory terms, based on the ANOVA with all nine wines, the panel could not detect significant differences at the 95% confidence level for any of the attributes, and there were no significant interactions among closures. Moreover, there were no significant single correlations between QDA attribute intensities within closure types based on browning levels: low, medium, and high (*p* ≤ 0.05).

However, when combining the responses to all the wines, for the analysis of the two main effects, oxygen level and closure type, only for the main effect oxygen level, significant differences were found for two attributes at the 95% confidence level, appearance in ‘golden color’, and in ‘burning’ aftertaste. The mean intensities of these two significant attributes are shown in Figure 3 by oxygen level. Not surprisingly, ‘golden color’ was found to be higher in wines with the highest level of browning. Interestingly, wines overall with a medium level of browning had a greater burning aftertaste.

## 3. Discussion

### 3.1. Browning Trajectories of a Sauvignon Blanc Wine under Three Different Closures and Withing Bottle Variation at 30 Months

The mean browning levels calculated with the quadratic growth function at 18 and 30 months showed the most similarities for synthetic closures and natural corks compared to screw caps. These browning trajectories indicated TPO differences at bottling for screw caps, most likely because of larger HS volume (Table 4). With comparable headspace oxygen concentration, this would have introduced more oxygen at bottling, oxygen that would have been consumed relatively quickly. Overall, the screw caps show higher browning mean values than natural cork and synthetic closures, starting from the earliest time point, at 4 months, and continuing throughout the 30 months. The difference can be explained due to their technical application properties. For screw cap, a head space larger than 30 mm is recommended for expansion during storage at 20–25 °C [30]. That additional headspace oxygen was no doubt responsible for the persistent 0.05 AU increment for the screw cap wines.

In general, inserted closure types have smaller and more similar HS at bottling compared to screw caps, but the increase in color at the beginning of the storage period is related to the release of oxygen that is contained in these inserted closures over those first few months. Oxygen in the material’s cavities degasses into the wine right after bottling due to oxygen partial pressure differences [10,13,25,28,31]. The similarities of the mean browning values especially at 18 months for cork and synthetic closures are very likely related to these material and technical application similarities at bottling.

Interestingly, the browning trajectory for natural cork had a significant deceleration at the final stages of the experiment (Figure 1). The natural cork-closed wine bottles were stored inverted, and the wine had a chance to be in contact with the closure during storage. It was possible to see that some natural corks absorbed wine into the cork matrix. It is probable that wine present in the cork reduced the oxygen ingress into the bottled wine, slowing the oxidation reactions, and consequently browning reactions. The actual explanation for this observation needs to be studied further, but Fonseca et al., 2013 [32] found that wetting cork by water or ethanol decreased gas permeability by 10-fold. Further, a study from Oliveira et al., 2015 [33] found that natural corks of low quality that had absorbed some wine had decreased OTR, similar to natural corks in the best quality class.

In addition, after 30 months of aging, the free SO_2_ values of tested wine bottles closed with natural cork were the closures with the highest free SO_2_ levels for the low (NL) and medium (NM) browned wines (Figure 2B), which supports the theory that cork material decreases oxygen ingress over time.

The natural cork, high (NH) browned wines were overall much lower in free SO_2_, and we saw a higher variation in the cork closures (Figure 2B and Figure 3), suggesting, not surprisingly, that natural materials have less consistency than manufactured products. The lower coefficient of variation of the screw cap closures and the synthetic closures can be explained by their manufactured nature; however, as reported in the literature, there is still substantial inconsistency, which could be caused by variation inherent in all physical objects and processes. These would include defects in the closure or the glassware, and variations in the application process [24,34].

For any closure, variation in filling height within a bottling run can result in different amounts of TPO. In this study, all bottles were filled in the middle of the bottling run to minimize this effect. Twenty randomly selected bottles from each 200-closure type bottle pool were measured for their headspace volume to check for a consistent filling after bottling within each closure type. The headspaces that are shown in Table 4 were overall consistent within closure types with small standard deviations, respectively.

### 3.2. Sensory Evaluation

The sensory evaluation was deferred until 30 months to increase the possible sensorial differences to test the closures barrier performance within closure treatments. The sensory impact of the closures on the wine was very similar across all closure types as described in the results. The QDA sensory panel used typical aroma and flavor sensory terms for Sauvignon blanc to describe all wines. Some of the fresh Sauvignon blanc aromas included ‘fruity’, ‘citrus’, and ‘green apple’, which also agreed with the winemaker’s notes. The trained panel did not note any oxidative aromas that would appear in oxidized wines (e.g., ‘bruised apple’, ‘honey’, ‘nutty’) or reductive aroma terms (e.g., ‘struck flint’, ‘rubber’) nor cork taint, and did not find any significant differences among wines for those fresh aroma terms within or among closure types. In previous studies where natural cork closures or screw cap closures were tested and compared [8,11,19]), often the sensory data that pointed out differences between closures belonged to those groups. 

While a variation in browning and therefore oxidation within closures was detected using a sensitive analytical instrument, the trained non-expert panelists could not perceive such differences within closure types. The visual ‘golden appearance’ of the wine was only apparent to the panelists when all closures were aggregated (Figure 4), which indicates that at 30 months of aging had a sensory impact on the wines and that color change could be measured by the human eye.

Even so, the QDA method effectively created a tasting profile for all wines and showed significant differences overall in browning. A consecutive step could have been to perform a discrimination test and ask a larger group of panelists for the significant term ‘golden appearance’ to test for the difference within the closure’s browning levels. Nevertheless, we would not have had enough sample bottles available to do such a powerful sensory test with a higher number of tasters. 

The lack of sensorial differences among and even between closure might be surprising, as well as the lacking vocabulary for aging-related aroma terms, and one might consider trained wine experts would have detected more aroma differences among wines. It might be true that wine expert tasters would have eventually chosen a different, more concise vocabulary from their long-term memory. However, a well-trained panel, no matter the specific knowledge background, is a very consistent and robust sensory tool to detect significant differences among wines [35,36].

Furthermore, a 10-year Sauvignon blanc wine closure study investigating oxidation aroma intensity of different OTR levels [37] showed that an oxidative sensory impact on the wines closed with closures in the OTR range 0.3, 0.4, 0.6, and 1.5 mg of O_2_/year became, especially for the three lower OTR closures, more apparent after a longer aging period. Coetzee et al., 2016 also found that Sauvignon blanc wines with different levels of total consumed oxygen (TCO), and by consequence, different free SO_2_ levels, showed visual browning changes before oxidative aroma shifts were noticeable [38].

Sacks et al. [39] noted that under conditions of fairly rapid oxidation (bag in box containers), the consumption of bound (measured as total after free SO_2_ levels went to zero) SO_2_ continued at the same rate as free, indicating that even bound SO_2_ functions as an antioxidant. Thus, if the wines had suffered little oxidation prior to bottling and had low levels of oxidation products, then there would be little to distinguish the low SO_2_ wines from an aroma perspective. As Sacks notes, while oxidation does not occur as long as there is some measurable total SO_2_ present, the lack of free SO_2_ allows for the release of constituent aldehydes previously bound by SO_2_. Under these circumstances, wines will then smell oxidized, but that is contingent on having such aldehydes present in the wine.

The similarities in the sensory profiles of our investigated wines could be generally explained by their levels in sulfur dioxide. In our experiment, free sulfur dioxide was still present in all wines at the study’s endpoint (Figure 2B). The natural cork high-browning (NH) group still had 5 mg/L of free sulfur dioxide. While it might be unexpected that a difference of 12 mg/L would be perceptible to a sensory panel, even the modest level of 5 mg/L appears to have suppressed oxidative aromas sufficiently for a consumer panel. This might be expected as the dissociation constant (Kd, M^−1^) between acetaldehyde and SO_2_ is relatively small (1.5 × 10^−6^) and would be expected to be similar for other oxidation products (i.e., Hexanal: 3.5 × 10^−6^). Thus, with even low levels of free SO_2_, there would be virtually no free aldehydes present, and they would have no impact on the aroma. This result does demonstrate that SO_2_ is an effective preservative even at low levels, a result that was predicted by Nikolantonaki et al., 2014 [40] who showed that SO_2_ reacted very quickly with wine oxidation quinones, making it a potent preservative even at low concentrations.

An 18-month bottle storage study of Argentinian Torrontes Riojano white wine confirms our results under different temperatures and similar closure choices. Under warmer, 25 °C storage conditions, oxidative sensory differences were detected among closures. However, no sensory differences were detected under the lower 15 °C temperature, where similarly to our study, the wines still had 5 mg/L of free SO_2_ available. In contrast, the higher-temperature storage wines had no free SO_2_ left [41].

Other studies suggested that oxidation aromas should have been noted in our wine sensorially with the lowest SO_2_ level. Godden et al., 2001 [19] indicated that 10 mg/L of free SO_2_ was the threshold for a high risk of advanced oxidation and color formation in wines containing free SO_2_ content. However, when zooming in on the AWRI closure study, the research team followed the sensory profile of the Semillon wines over three time points. After 6, 12, and 18 months, they performed a DA with trained expert wine tasters that generated possibly more precise terms from their long-term memory related to wine aging. They detected small differences among closures after 6 and 12 months, especially for a subset of closures, and the drivers were aromas related to musty, TCA, and oxidation. The research team decided to boost the statistical power over time, evaluating at last eight samples at 18 months at the last sampling. At this timepoint, significant sensory differences were more apparent among the 14 closures due to the true time aging effect and possibly the increased sampling size. Overall, the closure pool for this study was more diverse in OTR and material properties, which was consequently an additional driver for distinct differences. Three of their closures most comparable to our study in material and free SO_2_ content performed similarly in some fruit-related sensory terms. However, there were more differences between developed and oxidized and reduced aroma, where the latter depends on the wines’ bottling redox state.

Two studies, including a wine oxidation study from South Africa supports that an aroma shift of their Sauvignon blanc wine from fresh and fruity aroma profile to more oxidized aromas started to happen at 10 mg/L free SO_2_ [38]_._

Similarly, in 24-months aged Sauvignon blanc wines from New Zealand closed with a screw cap and natural cork, with SO_2_ levels as low as 9 mg/L for cork, a trend for an oxidative-related decrease in volatile thiol compounds was detected. However, the trained DA panel did not find a significant difference in fruity and green aroma terms yet and did not mention oxidation or reduction-related terms.

The studies discussed in the last paragraphs were Sauvignon blanc wines aside from the Australian Semillon. Even if most of these studies were researching the same variety, an individual wine matrix effect should always be considered. Due to the interactions of odorants and other non-volatile components in the wine, the sensory perception of aroma, taste, and mouthfeel can be overall obscured or enhanced and challenging to compare [42].

## 4. Materials and Methods

### 4.1. Wine and Winemaking

Sauvignon blanc from Napa, California, USA hand-harvested in September 2011 at 22.6°Brix. The final wine blend was 96% Sauvignon blanc, 2% Sauvignon Musque, and 2% Semillon.

The wine was made at CADE winery, crushed, and then pressed using a XPF40 bladder press (Bucher Vaslin, Chalonnes sur Loire, France). The pressed juice was pumped into a chilled stainless-steel tank and was allowed to settle overnight. After 24 h, the juice was racked off its gross lees and transferred to different fermenter tanks. The juice was inoculated in the following fermentation vessels with commercial yeast. For the final blend wine from 53% stainless steel tank fermentation, 20% stainless steel drum fermentation, 22%-barrel fermentation (French oak, acacia, and cigar barrels), and 5% egg-shape concrete tank fermentation was used. Yeast nutrients were added as needed to aid fermentation. Once the residual sugar was below 1.0 g/L, 30 mg/L of potassium metabisulfite was added to the wine. The wine was allowed to settle for 3 weeks. The wine in the stainless-steel tanks was racked off their gross lees and kept in tanks without headspace. The wine in oak barrels, stainless steel drum, and the egg-shaped concrete tank was aged sur lie. The wine was topped every 2–4weeks as needed to prevent headspace. The wine was blended in November 2011 and held in a full stainless-steel tank until bottling at 10 °C. Bentonite at approximately 5 lbs./1000 gallons was added to attain heat stability; although heat stability (temp) below 1 NTU was not reached, no more bentonite was added.

The wine was also tested for cold stabilization after the bentonite addition. The tank was chilled down to −1 °C and held at that temperature until stability was reached. A sample was analyzed using the methods of ETS to verify that cold stability was reached. The wine was racked off the bentonite lees and tartrate residue 2 days before bottling and sterile filtered. For the sterile filtration, an in-line set up with two filters of the 1900 Cellu-Stack cartridge series was used. All filters were obtained from Gusmer Enterprise, Inc.^®^ (Mountainside, NJ, USA). The pre-filter housing cartridge used was a 1945 pre-filter (1 μM nominal), and the second housing was a 1965 sterile filter (0.45 μM absolute). The wine was held for 2 days in a tank at 10 °C until bottling.

In March 2012, the wine was bottled with the following analytical parameters post-filtration: 36 mg/L of free SO_2_, 6.6 g/L of titratable acidity (FTIR), pH 3.43, glucose and fructose 0.6 g/L, ethanol (FTIR) 14.12% vol. One month after bottling, the free SO_2_ was 23 mg/L.

### 4.2. Bottles and Closures

For all closures, Bordeaux mold flint glass bottles from Owens-Illinois glass (O-I glass) were used. The screw cap bottles had the mold number W1511; the synthetic and natural cork bottles W1506. The glass was purchased from Demptos (Demptos Glass Company, Fairfield, CA, USA).

The bottles had a very shallow punt with a depth of 0.563 inches, so when the absorbance was measured in the spectrophotometer, there was no interference. Three different closures were used in the experiment, and for each closure type, 200 bottles were bottled. The natural corks were donated by Amorim (Amorim Cork America, Napa, CA, USA). The specifications were 49 × 24 mm CA-05 Bwc+.

Synthetic corks used were Nomacorc Select 300 series 44 × 23 mm, donated by Nomacorc (Cork Supply, Benicia, CA, USA). According to Nomacorc specifications, the oxygen ingress into wine bottles is as follows: 1.35 mg of O_2_ after 3 months, 1.79 mg of O_2_ after 6 months, 2.4 mg of O_2_ after 12 months, and 1.7 mg of O_2_ per year after the first year.

Screw caps donated from Amcor were 30 × 60 mm ROPP Stelvin capsules with Saranex liners (Amcor Flexibles Inc., American Canyon, CA, USA) with an average OTR of 0.5 mg/year [24,29].

### 4.3. Bottling

The wine was bottled in the middle of the wineries’ regular bottling run of the 2011 Napa Valley Sauvignon blanc to minimize filtration problems, fill heights, applied screw cap pressure, and inconsistency in total package oxygen. A mobile bottling line contractor performed the filling of the bottles (Ryan mobile bottling, Napa, CA, USA). The wine passed again through a sterile filter before entering the filler bowl. All bottles were inverted and blanked with winery-provided nitrogen, while the vacuum aided in drawing out possible particles in the bottles (US Bottlers DS8 Sparger). The wine was filled with a 4D machine 24-spout custom filler. The inserted closures were applied with a Bertolaso delta 604R corker and the screw caps with a Zalkin CA4 ROPP screw capper.

The screw cap bottles were bottled first according to the wineries’ commercial production of the wine. The bottles’ headspace was dosed with liquid nitrogen to displace any containing oxygen before the application of the screw caps using an ultradoser headspace injector. Second, the bottling line was prepared for natural corks and Nomacorc. All inserted closures were labeled with numbers and fed into the corking machine manually, so numbers were facing up visibly after corking. During the cylindrical closure application, a vacuum was pulled against the bottles to evacuate any oxygen in the headspace.

After bottling, all the 600 wine bottles were packed into boxes inside the winery’s cave. The wine boxes were transported to UC Davis, where they were stored in the dark and under room temperature-like conditions in the winery cellar. The average controlled temperature was 21.3 ± 0.9 °C.

The 200 bottles with natural cork were stored neck down to keep the cork moistened with wine, whereas the synthetic corks and the screw caps were stored neck up.

Average headspace measurements were taken on a sample size of 20 bottles for each closure type and can be found in Table 4.

### 4.4. Determination of Browning at 420 nm (a Proxy for Oxidation)

All 599 bottles were measured periodically over 30 months with a modified laboratory spectrophotometer (8454 UV-VIS with photodiode array system, Agilent Technologies, Santa Clara, CA, USA) at 420 nm, determining their levels of browning. The measurement time points (T) were: 0, 3.3, 8.2, 12.3, 16.2, 20.3, and 30 months. The wine appeared to be slightly cloudy the day after bottling (T0), and those measurements were not used for analysis as all bottles had lower absorbance readings after 3 months of aging. The method used for evaluating wine aging is a published in-situ method of evaluating the brown color of white wine in clear and colored bottles using a modified laboratory spectrophotometer [43]. All bottles were measured upright and placed directly into the beam of the spectrophotometer. Each bottle was marked at the bottom of its glass rim to easily line up each bottle in the same orientation for each subsequent measurement in the spectrophotometer. All measurements were taken twice and averaged for analysis. The spectrophotometer was zeroed against air before each set of data collection. Each bottle was used as its own control with respect to time.

### 4.5. Sulfur Dioxide Measurements

The wines that were sampled for the sensory evaluation were tested for free and total sulfur dioxide using flow injection analysis (ETS Laboratories, St. Helena, CA, USA) using a Foss FIAstar™ 5000 (Foss, MN, USA). The method is based on aeration-oxidation. Each day, bottles were opened for sensory analysis; a subsample was carefully transferred to a 60 mL centrifuge tube without headspace and kept cold until analysis was performed the following day. A total of five bottles for each closure type/browning level was measured.

### 4.6. Quantitative Descriptive Analysis (QDA)

At the endpoint of the study, at 30 months, a trained sensory panel evaluated the Sauvignon blanc wines. Within each closure type the research team selected three levels of browning to represent lower, medium, and higher oxidation levels (Table 3 for terminology) based on analytical measures at 420 nm (Figure 2A). The sensory panel described the Sauvignon blanc wine aged with the three different closures: natural cork, synthetic cork, and screw cap, using sensory descriptors chosen during their training based on consensus. In total, the panel evaluated nine wine treatments. The trained sensory panel was developed using the Tragon QDA method (Redwood City, CA, USA). The QDA panel consisted of 11 individuals that had been previously screened for their sensory acuity on white wines and trained in the QDA process. Under the direction of a trained Tragon QDA moderator, the panel developed a sensory language to fully describe the wines, along with the evaluation procedures. The language development process required six 90-min sessions held on consecutive weekdays. The panel, as a group, was trained on consensus sensory terms that were divided into five modalities: appearance, aroma, flavor/taste, mouthfeel, and aftertaste/aftereffects (Appendix B) [44]. After language development, the panelists rated each wine individually on each attribute intensity using a 15 cm unstructured graphic line scale with anchors that describe each sensory terms intensity from not present to very intense (Appendix B). The wines were evaluated using a complete balanced block serving order such that each product was served in each position an equal number of times. All wines were tasted blind in separate sensory booths, in sequential monadic order with 3-minute timed rest intervals between products. Each wine was evaluated in triplicate to provide a sufficient database for statistical analysis.

### 4.7. Sensory Data Analysis

For computation purposes, subjects’ sensory responses were recorded and analyzed using RedJade software suite (www.redjade.net (accessed on 27 October 2014).; Redwood City, CA, USA). Ratings on the 15 cm unstructured line scales were converted to numbers from 0 to 60 for data analysis. RedJade is specially designed to collect and analyze QDA and trained panel data. The analyses consisted of a one-way analysis of variance (ANOVA) for each sensory attribute to measure consistency in rating across panelists and contribution to identifying product differences. Further, a two-way ANOVA (treatment-by-subject with repeated measures) for each sensory attribute was applied to determine whether the panel scored the products as different from one another. After the ANOVA, the Duncan multiple range test was applied to identify statistically significant differences among products for each sensory attribute. Product means, standard deviations, and panelist ranking of each product for each sensory attribute were calculated.

### 4.8. Data Analyses of Browning Levels and Free and Total SO_2_ at 30 Months for Sensory Panel

For each closure group, 200 bottles were measured repeatedly over 2.5 years in a spectrophotometer at 420 nm, except for the natural cork group where, due to breakage, 199 bottles were measured.

At the end of the experiment, all repeated absorbance measurement readings from time points: 3.3-, 8.2-, 12.3-, 16.2-, 20.3-, and 30-month days were plotted against time. These data slope values calculated by linear regression in MS Excel 14.5 for each bottle became µ-absorbance reading units/day, a comparable measure of browning rate over 30 months of storage for each bottle.

The research team selected five bottles for sensory evaluation based on the slope values for each closure group: at the medium browning level and the high and low ends, avoiding the extremes (Figure 2A).

We tested the slope value, free and total sulfur dioxide data, for all selected bottles, for normal distribution and homogeneous variances, using Shapiro–Wilk and Levene’s tests, respectively. Subsequently, the slope value data were log2 transformed. For mean difference among browning, free, and total sulfur dioxide levels within each closure group, data were analyzed with a one-way ANOVA using Tukey’s honest significance test (Tukey HSD, *p* < 0.01) in R version 3.2.0 (Figure 2A–C).

### 4.9. Data Analysis of Browning to Estimate Variation within Closure Group

For each closure type, a coefficient of variation (*n*-1) of the slope values obtained above was calculated, and scatter plots (Figure 3) were created using XLSTAT, Addinsoft (2021) to visualize closure-related variation after 30 months of aging.

### 4.10. Data Analysis of Browning Trajectories within Closure Groups

To better comprehend the data, all absorbance readings, where six repeated measures of browning levels starting from 100 days to 30 months 914 days after bottling, were analyzed. Levels were analyzed according to 30-day increments (to estimate the change in approximately monthly increments). In the data analysis, we used the actual measures of time at which browning levels were taken and coded time to reflect 30-day increments (to estimate the change in approximately monthly increments). We defined time = 0 to represent browning levels at about 18 months, and in a separate model, at about 30 months after bottling.

Browning levels were analyzed using a random-effects regression model for longitudinal data. These models are commonly applied in the analysis of longitudinal data because they estimate the typical growth trajectory of a population. Besides, they characterize the extent to which experimental units vary in the features that characterize change [45]. For the browning data, a common function was assumed to describe the change in browning for all bottles over time. The coefficients of the growth function were allowed to vary across the individual bottle, which allowed for the study of both within-bottle variation in browning levels over time and between-bottle variation in the specific features that characterized the change in browning levels.

For each of the three closures, different functions were fitted to the browning levels to find a model that best approximates the change in browning levels across time. These growth functions included a model of no-growth, a linear growth model that assumes browning levels change at a constant rate, and a quadratic growth model that allows for a non-constant rate of change. These functions are described in the Appendix A. The estimation was carried out using maximum likelihood via PROC NLMIXED with SAS version 9.4. Within closure, model fit was evaluated using likelihood-based indices of fit, precisely the Akaike information criterion (AIC) and the Bayesian information criterion (BIC). The model with the smallest index values (AIC and BIC) was taken as the best fitting model (Table 1).

## 5. Conclusions

An oxidation proxy method, measuring absorbance at 420 nm, analyzed by a statistical random-mixed method for nonlinear data, characterized the storage performance of three common closure types, applied during a regular winery bottling over a 30-month room temperature storage period. In particular, the nonlinear behavior for natural cork closures suggests that the closure’s behavior changes over time to decrease oxygen transfer rates. It would be very interesting to evaluate these changes over an extended period of many years. The possibility that cork-wetting may be responsible for this evolution needs to be further investigated.

The low levels of free SO_2_ at 30 months, as little as 5 mg/L, appear to obscure oxidation products from trained non-expert panelists, making these wines hard to distinguish sensorially from the same wine that had retained more free SO_2_ and had much less browning. We would expect that when the free SO_2_ was depleted, consumers would detect aroma differences, so knowing the SO_2_ loss rate in bottled wine would be very important to judge when it might be prudent to pull the remaining bottles from the shelf. So, our winemaker’s projection that 24 months was a reasonable limit on shelf life would in fact ensure that consumers would not encounter an “off” bottle with a reasonable safety margin.

A good correlation between residual SO_2_ and browning rate showed that the browning in white wine is an excellent indicator of oxidation, even if non-expert tasters could not perceive sensorial differences.

With the assumption that at some low SO_2_ level oxidation would be perceived by consumers in a wine, the use of manufactured closures does reduce but not eliminate variability. It can give producers more confidence in projecting future SO_2_ levels for nearly all the bottles in a shipment, assuming reasonable storage conditions.

## Figures and Tables

**Figure 1 molecules-27-05881-f001:**
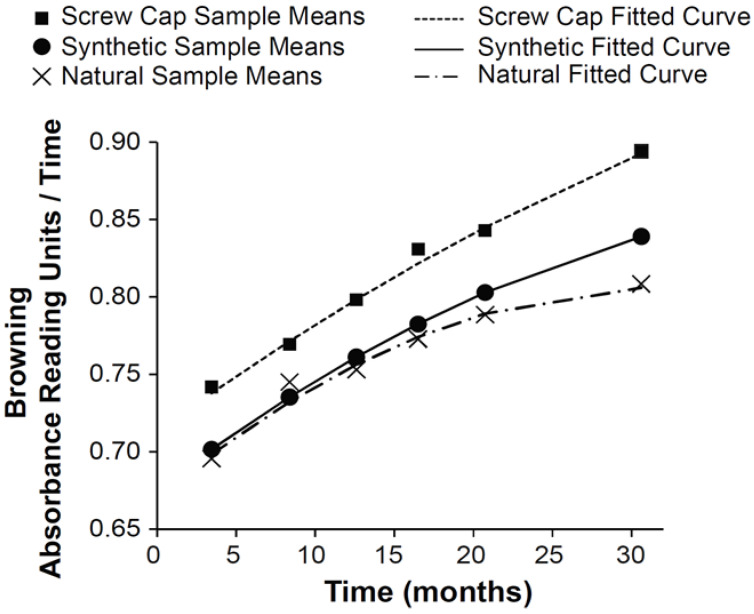
Fitted browning trajectories of the typical bottle by closure type.

**Figure 2 molecules-27-05881-f002:**
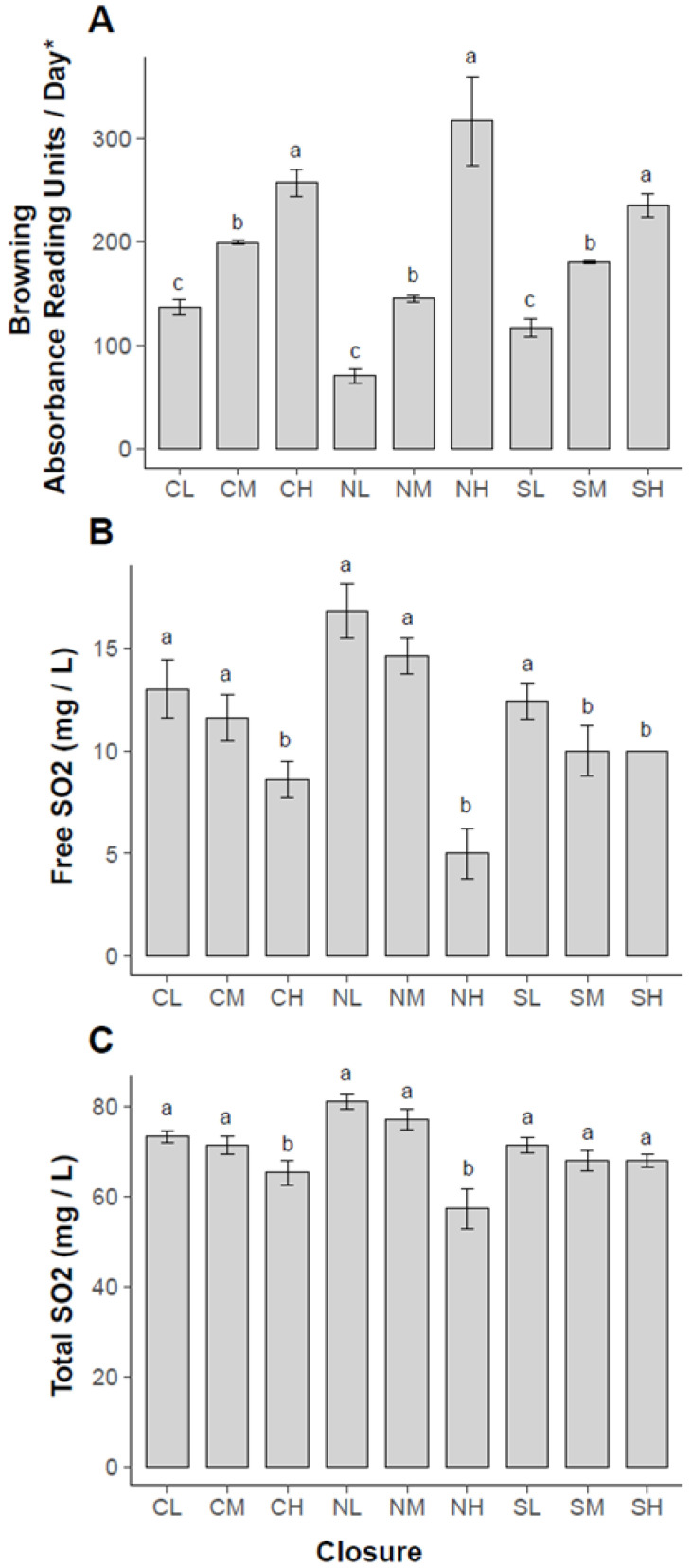
(**A**) Average browning (Abs 420 nm) (* all browning levels were magnified by 1 × 10^6^), (**B**) Free SO_2_,and (**C**) Total SO_2_ levels of bottles chosen for QDA sensory analysis and their significant differences within closure type. *n* = 5 bottles selected per closure/browning level; rows/bars sharing the same letter (a, b, or c) are not significantly different within closure group (*p* ≤ 0.05).

**Figure 3 molecules-27-05881-f003:**
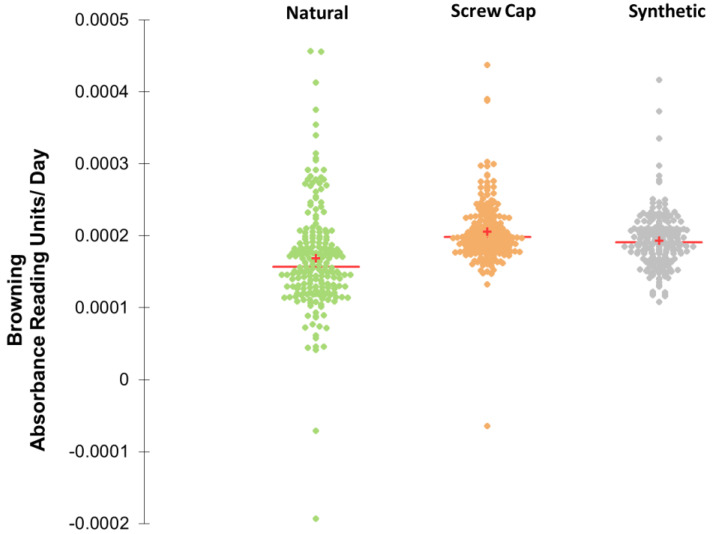
Scattergrams for linear browning μ-absorbance reading units per day for natural cork (green), screw cap (orange), and synthetics (grey). The red crosses correspond to the means and the central red horizontal bars represent the medians. The coefficient of variation (CV) for natural cork is 44.8%, for screw cap 20.6%, and for synthetic 21.9%.

**Figure 4 molecules-27-05881-f004:**
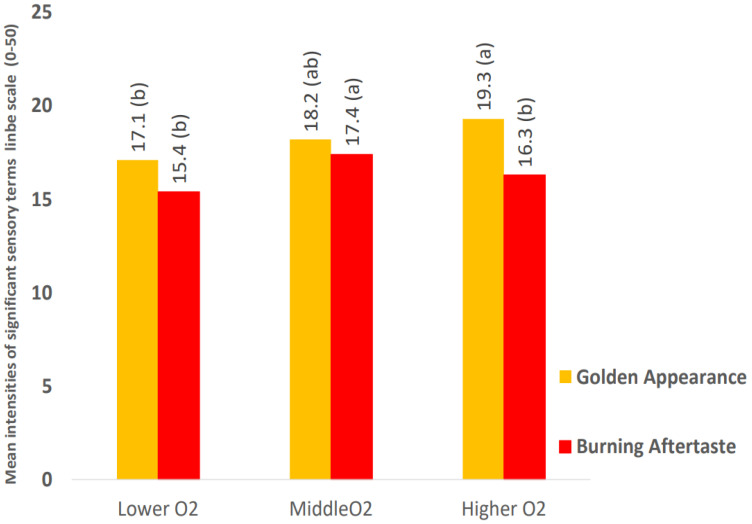
Mean intensities of significant sensory terms of main effect oxygen level. Note: Significance at *p* < 0.05, attribute means showing the same letter (a or b) are not significantly different.

**Table 1 molecules-27-05881-t001:** Indices of fit for growth models fit to browning level.

Closure	Fit Index	No Growth	Linear Growth	Quadratic Growth
Synthetic (*n* = 200)	−2lnL	−3101.6	−5031.9	−5152.0
	AIC	−3095.6	−5023.9	−5142.0
	BIC	−3085.7	−5010.7	−5125.5
Natural (*n* = 199)	−2lnL	−3262.3	−4099.0	−4214.7
	AIC	−3256.3	−4091.0	−4204.7
	BIC	−3246.4	−4077.8	−4188.2
Screw cap (*n* = 200)	−2lnL	−3011.7	−5583.4	−5646.5
	AIC	−3005.7	−5575.4	−5636.4
	BIC	−2995.8	−5562.3	−5620.0

Note: For fit indices, −2lnL is −2 times the log-likelihood; AIC is the Akaike information criterion; BIC is the Bayesian information criterion. All models include a random intercept.

**Table 2 molecules-27-05881-t002:** Maximum likelihood estimates (MLE) of the growth model parameters by closure.

Closure	Parameter	MLE (SE)	95% CI
Synthetic	β0, browning * at 18 months	0.79(0.006)	(0.78, 0.80)
	β0, browning at 30 months	0.84(0.006)	(0.83, 0.85)
	β1, linear change at 18 months	0.0049 (0.00006)	(0.0048, 0.0050)
	β1, linear change at 30 months	0.0030 (0.00019)	(0.0027, 0.0034)
	β2, acceleration rate	−0.00008(6.9 × 10^6^)	(−0.00009, −0.00006)
	φb0, between-bottle variance	0.0073 (0.0007)	
	σ2j, within-bottle variance	0.0003 (0.00002)	
Natural	β0, browning at 18 months	0.78 (0.005)	(0.77, 0.79)
	β0, browning at 30 months	0.81(0.005)	(0.80, 0.82)
	β1, linear change at 18 months	0.0037(0.00010)	(0.0035, 0.0039)
	β1, linear change at 30 months	0.0006(0.00031)	(0.00001, 0.00124)
	β2, acceleration rate	−0.00013 (0.00001)	(−0.00015, −0.00010)
	φb0, between-bottle variance	0.0052 (0.0005)	
	σ2, within-bottle variance	0.0009 (0.00004)	
Screw cap	β0, browning at 18 months	0.83 (0.005)	(0.82, 0.84)
	β0, browning at 30 months	0.89 (0.005)	(0.88, 0.90)
	β1, linear change at 18 months	0.0056 (0.00005)	(0.0055, 0.0057)
	β1, linear change at 30 months	0.0046 (0.00015)	(0.0043, 0.0049)
	β2, acceleration rate	−0.00005 (5.6 × 10^6^)	(−0.00006, −0.00003)
	φb0, between-bottle variance	0.0049 (0.0005)	
	σ2, within-bottle variance	0.0002 (1.0 × 10^5^)	

Note: * Browning levels over time fitted by a quadratic growth function with a random intercept. Standard errors are in parentheses. 95% CI are estimated 95% confidence intervals.

**Table 3 molecules-27-05881-t003:** Closures evaluated in QDA.

Closure Type and Oxidation/Browning Level
Screw cap Low O_2_ (CL)
Screw cap Middle O_2_ (CM)
Screw cap High O2 (CH)
Natural cork Low O_2_ (NL) Natural cork Middle O_2_ (NM) Natural cork High O_2_ (NH)
Synthetic cork Low O_2_ (SL)
Synthetic cork Middle O_2_ (SM)
Synthetic cork High O_2_ (SH)

**Table 4 molecules-27-05881-t004:** Average headspace measurements in cm and standard deviation for each closure type (*n* = 20).

	Screw Cap	Synthetic Cork	Natural Cork
HS (cm)	4.8	1.5	1.3
S.D.	0.1	0.2	0.1

## Data Availability

Samples of the compounds are not available from the authors.

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
