# Peer review of "Wine Closure Performance of Three Common Closure Types: Chemical and Sensory Impact on a Sauvignon Blanc Wine"

_molecules, 2022, doi:10.3390/molecules27185881_

Round 1

Reviewer 1 Report

The paper investigated the browning dynamics of a bottled Sauvignon Blanc wine under 3 types of closures over 30 months’ storage, combing with sulfur dioxide content and a brief sensory analysis after the last sampling, and discussed the correlation between them mentioned above in detail based on wine closure consistency. It impressed me much in dealing with the browning trajectories of Sauvignon Blanc wine under 3 closures. However, except sulfur dioxide, few other important chemicals (e. g. phenolic compounds) related to wine browning were involved. As well, sensory analysis of QDA data by multiple statistical analysis (e.g. PCA) is not enough in the paper.

Main comments:

  1. The browning trajectories of Sauvignon Blanc wine under 3 different closures are well written, while just free and total sulfur dioxide were dealt with at 30 months’ storage, how about other chemical parameters and possible statistical differences between 3 closures, such as basic chemical parameters (e.g. pH, volatile acid, etc.), phenolic compounds(e.g. total phenolics, individual phonics, etc.), even oxygen consumption rate and so on, so as to understand the chemical impact of closure consistency on Sauvignon Blanc wine in total, in accordance with the Title of the paper.

  1. QDA sensory analysis of different wine samples at 30 months was done by a tasting panel, more are dealt with a few texts and table 3, while no figure of flavor profile is provided in the manuscript (line 342?), although a detail discussion is offered in “1.2. Sensory evaluation”. Plots of multiple statistical analysis (e.g. principal component analysis) can be a useful tool to directly illustrate the sensory effect of different closures on Sauvignon Blanc wine. In particular, utilizing the flavor terms selected according to “4.6. Quantitative Descriptive Analysis” and table 3. However, neither the final selected terms based on table 3 nor their intensities are provided in the part (lines 175-183, lines 184-189), for just a brief discussion (lines 342-354) is not enough to demonstrate the sensory effect of closures.

  1. The numbers of tables or figures are not in order of their presence in the manuscript, which always make me confused. Table 3 seems long and duplicated, particularly the definition is not so specific, and it is better to delete “from weak to strong” from the table and quantify them in “4.6. Quantitative Descriptive Analysis” (line 537). Figure 4 is surplus to some extent, delete?

  1. For references, few literature cited are within the recent 3 years.

Specific comments

Line 92, Figure 2? The first figure present in the manuscript should numbered Figure 1? it is better to display the figures in order.

Line 96, Table 5, should be Table 1?

The same to the below. Line 102, line 104, line 118, line 121…

Lines 133-134, “2.2 Results of browning free and total sulfur dioxide concentrations after 30 months of aging and for QDATM sensory analysis”, delete “browning” and “for” in the phase.

Line 139, μ-absorbance units?

Lines 149-153, low, medium, high end of the browning scale, combing with lines 522-523, it is interesting. What is the specific absorbance value range or average for each group in the manuscript? What about the similar classification for browning in literature?

Line 165, delete the second “and” in the sentence.

Line 169, delete “Figure 4”? for more text mentioned above.

Lines 536-537, what’s unstructured graphic line scale? e.g. 15 cm linear scale? what’s the relationship with the value obtained in lines 545-546? 

Author Response

I highlighted all my text changes in yellow throughout the manuscript.

Reviewer comments in black font. 

Author responses in red fond.

I highlighted all my text changes in yellow throughout the manuscript.

The paper investigated the browning dynamics of a bottled Sauvignon Blanc wine under 3 types of closures over 30 months’ storage, combing with sulfur dioxide content and a brief sensory analysis after the last sampling, and discussed the correlation between them mentioned above in detail based on wine closure consistency. It impressed me much in dealing with the browning trajectories of Sauvignon Blanc wine under 3 closures. However, except sulfur dioxide, few other important chemicals (e. g. phenolic compounds) related to wine browning were involved.

The sensory analysis resulted in only two significant terms when combining all responses for the main effect of oxidation for all closures. A multivariate approach like PCA is appropriate only with having at least 5+ significant sensory terms and more products. I included a histogram showing mean intensities for the two significant sensory terms for the three oxygen levels (Figure 4) to visualize the differences.

Main comments:

  1. The browning trajectories of Sauvignon Blanc wine under 3 different closures are well written, while just free and total sulfur dioxide were dealt with at 30 months’ storage, how about other chemical parameters and possible statistical differences between 3 closures, such as basic chemical parameters (e.g. pH, volatile acid, etc.), phenolic compounds(e.g. total phenolics, individual phonics, etc.), even oxygen consumption rate and so on, so as to understand the chemical impact of closure consistency on Sauvignon Blanc wine in total, in accordance with the Title of the paper.

The analysis of other important chemicals is an excellent idea and important tool to assess oxidation. It would be a next step to investigate but far beyond the scope of what we had in mind. 

  1. QDA sensory analysis of different wine samples at 30 months was done by a tasting panel, more are dealt with a few texts and table 3, while no figure of flavor profile is provided in the manuscript (line 342?), although a detail discussion is offered in “1.2. Sensory evaluation”. Plots of multiple statistical analysis (e.g. principal component analysis) can be a useful tool to directly illustrate the sensory effect of different closures on Sauvignon Blanc wine. In particular, utilizing the flavor terms selected according to “4.6. Quantitative Descriptive Analysis” and table 3. However, neither the final selected terms based on table 3 nor their intensities are provided in the part (lines 175-183, lines 184-189), for just a brief discussion (lines 342-354) is not enough to demonstrate the sensory effect of closures

The sensory analysis resulted in only two significant terms when combining all responses for the main effect of oxidation for all closures. A multivariate approach like PCA is appropriate only with having at least 5+ significant sensory terms and more products. I included a histogram showing mean intensities for the two significant sensory terms for the three oxygen levels (Figure 4) to visualize the differences.

I added the consent sensory terms in the result section in the text. The table with the Panel QDA™ definitions of sensory terms for the Sauvignon blanc wines at 30 months of aging was moved to the appendix.

  1. The numbers of tables or figures are not in order of their presence in the manuscript, which always make me confused. Table 3 seems long and duplicated, particularly the definition is not so specific, and it is better to delete “from weak to strong” from the table and quantify them in “4.6. Quantitative Descriptive Analysis” (line 537). Figure 4 is surplus to some extent, delete?

I arranged the tables and figures and put them close to text they are mentioned first. I renumbered everything. The old numbered: "Table 3. Panel QDA™ definitions of sensory terms for the Sauvignon blanc wines at 30 months of aging" was moved into the appendix as Appendix 1. It explains the sensory terms and usually reference standards recipes or in the case of QDA refence standard recipes are explained in some part of a manuscript. The two only significant mean intensities were graphed in a histogram for the effect of oxidation on browning overall (all closure responses considered).  

I deleted the surplus Figure 4:  "Free and Total SO2 levels in high, medium, and low browning rate bottles selected for tasting trials" and put the reference to Figure 2A,B,C instead. 

  1. For references, few literature cited are within the recent 3 years.

I added a reference and text in the discussion line 352-357. Reference [42].

Specific comments

Line 92, Figure 2? The first figure present in the manuscript should numbered Figure 1? it is better to display the figures in order.

I changed the numbering as mentioned.

Line 96, Table 5, should be Table 1?

Was changed too.

The same to the below. Line 102, line 104, line 118, line 121…

Also changed.

Lines 133-134, “2.2 Results of browning free and total sulfur dioxide concentrations after 30 months of aging and for QDATM sensory analysis”, delete “browning” and “for” in the phase.

I did.

Line 139, μ-absorbance units?

Added explanation in the manuscript.

Lines 149-153, low, medium, high end of the browning scale, combing with lines 522-523, it is interesting. What is the specific absorbance value range or average for each group in the manuscript? What about the similar classification for browning in literature?

I added the specific ranges in the text.

Line 165, delete the second “and” in the sentence.

Done.

Line 169, delete “Figure 4”? for more text mentioned above.

I deleted it and replaced referencing to Figure 2A,B,C

Lines 536-537, what’s unstructured graphic line scale? e.g. 15 cm linear scale? what’s the relationship with the value obtained in lines 545-546? 

I added 15 cm line scale. An unstructured line scale does not have any measure ticks and is anchored at both ends with e.g. intensity rating terminology.

Final general author response: I highlighted all my text changes that address your suggestions in yellow in the manuscript.

Reviewer 2 Report

As it stands now, the manuscript reads more like a report with some exceptions (the discussion). For some reason, all the tables and figures are together under one subheading, which makes little sense and makes the reading difficult having to jump back and forth to find them. The Results section is extremely long and mostly repeats what can be seen in the tables and figures.

The authors also do not address the limitations of their work, and some of those limitations can have a large impact on the findings and the relevance of the work:

  • dissolved oxygen was not measured. the authors rely on the values from Nomacorc for the synthetic corks but how about the rest? looking at the varied values for browning, one possible explanation is different levels of oxygen ingress
  • derived from this, the authors chose three levels of browning for each closure, indicating inconsistency in this observed phenomenon
  • the conclusion from the sensory is that there were differences between the browning levels with some more detailed discussion for the closures but since each closure gave varied levels of browning, how is this discussion relevant in the bigger picture
  • flint coloured bottles are not the most typical for white wine, regardless of the reason why the authors chose them - so how would this work translate into practice?

Author Response

I highlighted all my text changes in yellow throughout the manuscript.

Reviewer comments in black font. 

Author responses in red fond.

As it stands now, the manuscript reads more like a report with some exceptions (the discussion). For some reason, all the tables and figures are together under one subheading, which makes little sense and makes the reading difficult having to jump back and forth to find them. The Results section is extremely long and mostly repeats what can be seen in the tables and figures.

I deleted the old figure 4 to shorten the results.

I fixed the issue with the subheading; tables and figures should be in order now and renumbered. They were placed close to the text they were mentioned first.

Please advise on shortening the results if still necessary.

The authors also do not address the limitations of their work, and some of those limitations can have a large impact on the findings and the relevance of the work:

  • dissolved oxygen was not measured. the authors rely on the values from Nomacorc for the synthetic corks but how about the rest? looking at the varied values for browning, one possible explanation is different levels of oxygen ingress.

We measured the dissolved oxygen at the beginning of the trial, but dissolved oxygen is consumed after a few weeks in wine and is not a good measure to follow oxidation progress over time. 

We are citing values for manufactured closures. Natural corks are highly variable, and each batch needs to be tested. 

The study aimed to assess the differences that the closure can cause; hence we chose to measure browning at 420 nm, which is an excellent proxy for oxygen ingress or oxidation. 

  • derived from this, the authors chose three levels of browning for each closure, indicating inconsistency in this observed phenomenon

Yes, we decided as a research team to explore the differences in browning within closure types based on statistical, significant browning values and if they would have a sensory impact. We clarified closure differences in the discussion as well.

  • the conclusion from the sensory is that there were differences between the browning levels with some more detailed discussion for the closures but since each closure gave varied levels of browning, how is this discussion relevant in the bigger picture

There were differences in browning by spectrophotometer, a very sensitive tool to detect oxidative differences; however, the panelist could not perceive this difference sensorially using our applied methods.  

I am not sure where you are heading with this comment. Please kindly clarify.

  • flint coloured bottles are not the most typical for white wine, regardless of the reason why the authors chose them - so how would this work translate into practice?

For the research study, and we needed to choose those bottles for technical applicability purposes; however, the bottles were stored in the cellar in dark boxes and only taken out for measurements as explained in the material and method part.

Round 2

Reviewer 1 Report

No more suggestions.